# The Application of a Hybrid Method for the Identification of Elastic–Plastic Material Parameters

**DOI:** 10.3390/ma15124139

**Published:** 2022-06-10

**Authors:** Beata Potrzeszcz-Sut, Agnieszka Dudzik

**Affiliations:** Department of Mechanics, Metal Structures and Computer Methods, Faculty of Civil Engineering and Architecture, Kielce University of Technology, al. Tysiąclecia Państwa Polskiego 7, 25-314 Kielce, Poland; agad@tu.kielce.pl

**Keywords:** parameter identification of material model, inverse analysis, indentation test, indentation curve, imprint profile, artificial neural networks

## Abstract

The indentation test is a popular method for the investigation of the mechanical properties of materials. The technique, which combines traditional indentation tests with mapping the shape of the imprint, provides more data describing the material parameters. In this paper, such methodology is employed for estimating the selected material parameters described by Ramberg–Osgood’s law, i.e., Young’s modulus, the yield point, and the material hardening exponent. Two combined identification methods were used: the *P-A* procedure, in which the material parameters are identified on the basis of the coordinates of the indentation curves, and the *P-C* procedure, which uses the coordinates describing the imprint profile. The inverse problem was solved by neural networks. The results of numerical indentation tests—pairs of coordinates describing the indentation curves and imprint profiles—were used as input data for the networks. In order to reduce the size of the input vector, a simple and effective method of approximating the branches of the curves was proposed. In the Results Section, we show the performance of the approximation as a data reduction mechanism on a synthetic dataset. The sparse model generated by the presented approach is also shown to efficiently reconstruct the data while minimizing error in the prediction of the mentioned material parameters. Our approach appeared to consistently provide better performance on the testing datasets with considerably easier computation than the principal component analysis compression results available in the literature.

## 1. Introduction

It is necessary to frequently inspect the mechanical properties of materials that potentially deteriorate due to aging processes under difficult environmental conditions. When the data of material properties are not available, the structure safety has to be verified using the limit state criteria. Adopting the appropriate material data for structure analysis requires the identification of the material parameters at many points of the system. Material identification problems are present, e.g., in the pipelines used for transporting hydrocarbons, which are the structural elements of the marine industry and power distribution companies. In all of these cases, the aim is to formulate diagnostic procedures that are non-destructive or almost non-destructive, fast, and economical [1]. Hardness testing is one of these methods.

The hardness test, also known as the indentation test, is a very common strength test used to determine the properties of the constitutive material parameters of materials such as steel, rocks, laminates, or even coffee beans. The popularity of the test is due to the simple measuring device (hardness meter) as well as the simplicity and speed of the measurements. Indentation is an almost non-invasive test that can be easily carried out in situ directly on the structural component. Many articles present an evaluation of mechanical properties such as carbon steel [2], bearing steel [3], austenitic stainless steel [4], and pipeline steel [5]. Paper [6] describes an experimental exploration of the post-impact behavior of pseudo-ductile carbon laminates. Article [7] incorporates the indentation test method into a geotechnical practice. An interesting approach is presented paper [8], where the authors used an artificial neural network (ANN) model to predict the plastic anisotropy properties of sheet metal. Moreover, an ANN combined with finite element analysis (FEA) might be used to derive a uniaxial tensile flow from spherical indentation data [9].

The subject literature [10,11,12] is devoted mainly to the assessment of material parameters resulting from the analysis of indentation curves. These curves show the relationships between the penetration depth of the indenter tip and the pressure force during sample loading and unloading. On the one hand, it is obvious that it is necessary to know these characteristics. On the other hand, however, it is not always possible to measure these parameters directly on the structure. This problem is solved by the use of the so-called hybrid methods.

The development of computer technologies in the last decades has allowed for the application of advanced numerical computations in many problems of science. Research is being carried out on computing using hybrid systems. One of the precursors to hybrid processing is Noor. The papers [13,14] present numerous applications of hybrid systems for the analysis of various problems in mechanics and materials. Computation hybridization is a combination of different computational methodologies, such as standard computing, called hard (HC) and soft computing (SC) [15,16], for the representation and processing of information. In practice, each type of calculation (HC, SC) has some advantages and limitations. Hence, the combination of different types of methods used in hybrid systems enables a reduction in the difficulties typical for each of them and makes it possible to complement each other and more fully use the strengths of both approaches [17,18]. In hybrid applications, SC, used for control, pattern recognition, signal processing, and identification, is hidden inside the HC computing systems or subsystems. SC can complement or replace HC to eliminate its limitations. Another use of SC is to create more user-friendly software features that could not be accomplished with standard computing alone.

The fusion of SC and standard computations has become a method of analysis used also by practicing engineers. In engineering, intelligent hybrid systems are used as elements that support the design of metal processing technology [19]. In recent years, hybrid systems have started to be used to solve complex problems of structural mechanics and materials. Some of the most effective computational techniques are numerical–neural hybrid systems (NNs) based on SC and standard computations [17]. Examples of the use of hybrid systems based on HC and NN in the field of engineering are the identification of a wide class of materials, including composites [20], soils [21], and geomaterials [22]. The integration of these methods may be used to formulate neural models of various types of materials, e.g., elastic–plastic or orthotropic [17]. The identification of the parameters of another model of an elastic–plastic material (Johnson–Cook) using data from hybrid bending experiments and their numerical simulations is shown in [23].

Systems integrating HC and NN are particularly useful in situations where good analytical models are unknown or very complex. In cases where analytical models are unknown or incomplete, NN can be used to formulate or estimate analytical models using experimental data. Using HC and NN is very popular in the case of material parameter identification [24,25,26].

In this article, we propose the performance of an approximation as a data reduction mechanism on a synthetic dataset. The sparse model generated by the presented approach is also shown to efficiently reconstruct the data,] while minimizing errors in prediction. Our approach is shown to consistently provide a better performance on the testing datasets with considerably easier computation than the principal component analysis (PCA) procedure, which has been proposed in the references [1,27].

## 2. Materials and Methods

### 2.1. Methods of Identifying Material Parameters

In practice, according to [27], the following procedures for identifying material parameters are applied:*P-A*: In situ hardness testers with the appropriate research software used for determining indentation curves. These curves are transferred to the computer, and parameter identification is performed based on this.*P-B*: The *P-A* procedure is supplemented with a laser profiler, which provides data on the imprinted shape. The imprint profile and indentation profile are both applied in the identification process.*P-C*: The third procedure is based on reading the coordinates of the imprint profile obtained by using a manual hardness tester. These data are input for inverse analysis.

The indentation test also generates some errors due to the fact that several materials with different yield strength values and strain hardening indexes may give identical force–displacement relationships. One way to overcome this problem is to use the double indent technique [28,29]. In such cases, it is also worth using identification on the basis of a double result of the indentation test, which is a combination of the *P-A* procedure and the *P-C* procedure. This approach allows for increasing the sensitivity of the identified parameters. It should be taken into account that the indentation curves generated during loading and unloading are susceptible to measuring system and data transfer errors, but the geometry of the imprint curves is usually reproduced correctly. An indentation test mentioned in this manuscript is based on the traditional Rockwell test (A and C scale), see [30]. The Rockwell hardness test is based on forcing a cone-shaped diamond indenter in the sample with hardness within the ranges provided by the A, C, D, and N scales or a ball-shaped steel indenter in the sample with hardness within the ranges provided by the B, E, F, G, H, K, and T scales. The test conditions are defined by the standard [30], the requirements of the hardness testers [31], and the standard of the calibration of hardness patterns [32]. Based on the indentation curves and imprint profiles, it is possible to obtain the values of the selected material parameters, e.g., the values of Young’s modulus, the yield point, and the material hardening parameter.

### 2.2. The Application of NN to Analyze Regression and Identification Problems

Due to its basic feature, which is the ability to generalize knowledge for new, previously unknown data, NNs are widely used in various scientific and engineering fields. Comprehensive reviews of the use of NNs in civil engineering have been presented in the literature, e.g., see [33,34,35,36,37,38]. In addition, networks can be formulated to analyze different regression problems by providing input/output data according to Paez’s classification [39]. In this study, NNs were used to analyze regression and identification problems. Solving a regression problem is related to estimating output values **y**(**x**;**w**) based on variable input vectors **x**:(1)$$x\to \mathrm{SN}\to y\left(x,w\right)$$
where **x**, **y**—network input and output vectors; **w**—vector of generalized network weights.

Inverse problems [34] concern situations where the system responses are known but there is not complete information on the reasons for this phenomenon. The inverse analysis combines experimental mechanical engineering with computer simulation and mathematical programming. In the first stage of this procedure, tests are performed from which measurable quantities are selected. In the next stage, experience is simulated. Then, mathematical programming is applied to reduce the objective function, which defines divergences between the value measured and its calculated equivalent [35]. In engineering, inverse problems are frequently implemented as a sequence of direct problems. In the case of more complex tasks, where both direct and inverse problems are considered, hybrid calculation systems are used, see [17].

This manuscript addresses a problem called internal identification, which corresponds to the identification of the parameters of the system material. Variable inputs act as the action parameters (applied force) and response parameters (displacement). Variable outputs are the material constants. A hybrid approach combining two computational models—FEM and NN—was applied to solve the identification problem. The material model is described by the Ramberg–Osgood (R–O) power law in Section 2.3. 

### 2.3. Ramberg–Osgood Material Model

This study used the so-called continuous model (curvilinear) that describes the relationship of σ(*ε*), assuming the form of a smooth curve, which reflects the material behavior in the elastic and elasto-plastic range with the potential reinforcement. Such a model can be applied to materials without a clear yield point, such as stainless steel [40] or aluminum [41]. Metals of that type, such as common hot-rolled structural steels, can also be modeled for design purposes as perfectly elastic–plastic.

In practice, however, it is necessary to perform more detailed modeling of material relationships. For this purpose, an R–O power law is often followed. Article [42] suggested a non-linear relationship between stress and strain in the following form:(2)$$\varepsilon =\frac{\sigma }{E_{0}}+p\;\cdot {\left(\frac{\sigma }{\sigma_{p}}\right)}^{n}$$
where *E*_0_—initial Young’s modulus; *σ*_p_—plastic strain for the adopted elastic limit; *n*—parameter characterizing material strengthening degree.

Equation (2) was primarily designed for aluminum steels, but it proved to be applicable to other metals with non-linear relationships, including stainless steel alloys. For design purposes, a conventional elastic limit $\sigma_{0.2}\;$ for a permanent strain $\varepsilon_{0.2}=0.2\%$ can be adopted as the elastic limit. With the above-mentioned assumptions, Formula (2) is as follows:(3)$$\varepsilon =\frac{\sigma }{E_{0}}+0.002\;\cdot {\left(\frac{\sigma }{\sigma_{0.2}}\right)}^{n}\;.$$

In order to estimate the value of exponent *n*, the reference stress level should be determined $\sigma_{x}$. For a known strain value $\varepsilon_{0,x}$ corresponding to stress $\sigma_{x}$, exponent *n* is given by the following formula:(4)$$n=\frac{\ln\left(\;\varepsilon_{0.2}/\varepsilon_{0,x}\right)}{\ln\left(\;\sigma_{0.2}/\sigma_{x}\right)}.$$

The coordinates of the second reference point $\left(\sigma_{x},\varepsilon_{0,x}\right)$ are determined based on the existing strains (stresses), considering the two following cases:
If the analysis is conducted within the elastic range (for $\sigma \leq \sigma_{0.2}$), then the stress corresponding to the permanent strain 0.1% can be assumed as a second reference point.If the analysis is conducted within the plastic range (for $\sigma >\sigma_{0.2}$), then the tensile strength corresponding to the highest point of the curve can be assumed as a second reference point.


For aluminum, the material strengthening degree is adopted from the range n∈⟨5, 48⟩ depending on the type and variety of steel and product type (sheets, tapes, sections, pipes, etc.) [41]. For stainless steels, parameter *n* is taken from the range n∈⟨4, 9⟩ within the standards [40,43,44]. In the task presented below, the model parameters are identified, for which isotropic elastic–plastic material with exponential amplification is described by the R–O law, see. [42], also known as Hollomon power law was assumed [45]. The originally formulated law for the uniaxial state is as follows:(5)$$\sigma \;\;=\;\;\left\{\begin{array}{ccc} \;E_{0}\;\varepsilon & \;\;\;\;\;\;\mathrm{dla} & \;\;\;\;\;\;\varepsilon \leq \frac{\sigma_{Y}}{E_{0}},\\ \;\sigma_{Y}\;{\left(\frac{E_{0}\;\varepsilon }{\sigma_{Y}}\right)}^{n} & \;\;\;\;\;\;\mathrm{dla} & \;\;\;\;\;\;\varepsilon >\frac{\sigma_{Y}}{E_{0}}, \end{array}\right..$$
where $\;\sigma_{Y}$—plastic limit in a uniaxial stress state.

Tri-axis version of this popular model is implemented in many commercial FEM programs, e.g., Abaqus [46]. 

Figure 1 shows a graphic interpretation of the R–O material model (5), showing the power relationship between strain and stress $\sigma \left(\varepsilon \right)$ for elastic–plastic material with given values: *E*_0_ = 200 GPa, $\sigma_{Y}$ = 380 MPa, and *n* = 0.092 [1].

The R–O law can also be applied to the modeling of structure material exposed to low-cycle loads. Articles [17,47,48] adopted a modified R–O model, where strain $\varepsilon \left(\sigma \right)$ during loading is defined by a skeleton curve (Figure 2):(6)$$\varepsilon =\frac{\sigma }{E_{0}}+\frac{2\sigma_{Y}}{3E_{0}}{\left(\frac{\sigma }{\sigma_{Y}}\right)}^{n}$$

However, the next loading/unloading cycles are described by a family of hysteresis loops, which are illustrated in Figure 2 (see [49]):(7)$$\varepsilon -\varepsilon_{R}^{e}=\frac{\sigma -\sigma_{R}^{e}}{E_{0}}+\frac{4\;\sigma_{Y}}{3\;E_{0}}\;{\left(\frac{\sigma -\sigma_{R}^{e}}{2\;\sigma_{Y}}\right)}^{n}$$
where $\;\varepsilon_{R}^{e}$, $\sigma_{R}^{e}$—strain and stress during unloading.

The skeleton curve is marked in black in Figure 2—it carries out the first cycle of loading in the mechanical system. The blue color marks the branch of the loop that unloads the system in each cycle. The branch of the loop that unloads the system in cycles from the second to the final one, depending on the needs, is marked in magenta.

### 2.4. Samples Generation by FEM

In papers [1,49], identification procedures of material parameters were performed. The research groups made the simulation results available for further analysis. The tasks were conducted by indentation tests using FEM for different material sets. The FEM is one of the basic tools of computer-aided scientific research and engineering analysis, with a very wide range of applications and high popularity. The possibility of using FEM requires knowledge from various fields of application (departments of physics, structural mechanics, etc.), mathematical knowledge about the basics, and IT knowledge to be implemented on computer hardware. Numerical computations were carried out in the ABAQUS [46] environment, which enables individual material characteristics to be used in calculations together with the generation of complex calculation grids and finite sliding formulation. 

In the program, the indentation test for a material sample 1 mm high and 2 mm in diameter was modeled. For the material of the test sample, the deformation plasticity as the material behavior was adopted. This model is primarily intended for use in developing fully plastic solutions in ductile metals. It does not need to be combined with any other mechanical material models since it completely describes the response of the material. The analyzed element was divided into 1690 four-node finite elements of various dimensions. The values and intervals between the lower and upper data limits assumed for the computations were as follows:
Young’s Modulus    170 GPa ≤ *E* ≤ 220 GPa      9 parts;Yield stress       330 MPa ≤ $\sigma_{Y}$ ≤ 460 MPa   13 parts;Exponent       0 ≤ *n* ≤ 0.2          40 parts.

Additional material parameters with constant values were Poisson’s ratio ν = 0.3 and the yield offset $\sigma_{0.2}$ = 100 MPa. The parameters of the mathematical model were identified based on the sample with a static diagram shown in Figure 3. 

The geometry and indenter tip’s material were compliant with the requirements of the standard [31]. A diamond indenter (Figure 4) with an opening angle of 120° and spherically truncated at a radius of 200 μm was adopted. An isotropic, linear elastic material was applied for the indenter. Constant elasticity values of *E* = 1140 GPa and ν = 0.07 were assumed, according to paper [50]. The indenter computational model was discretized by 144 finite elements (Figure 3).

Contact between the indenter and material sample was modeled using a pure master-slave contact algorithm: nodes on one surface—the slave—cannot penetrate the segments that make up the other surface—the master (Figure 3). After performing the computations, S = 4680 imprint profiles and 190 indentation curves were obtained, which had 36 and 100 points, respectively.

## 3. Case Study

In studies [50], to reduce the number of data describing both curves, principal component analysis (PCA) was used. PCA is also common in signal processing as discrete Karhunen–Loève transform (KLT) or a proper orthogonal decomposition (POD) in mechanical engineering, e.g., [49]. Owing to the reduced vector of input data, the assessment of the selected parameters (*E*, $\sigma_{Y}$, *n*) of elastic–plastic R–O material was relatively easy using NN. However, the PCA method has some disadvantages with the dimensionality reduction of datasets: it may lead to some amount of data loss, it tends to find linear correlations between variables, which is sometimes undesirable, and it fails in cases where the mean and covariance are not enough to define the datasets, see e.g., [51]. Therefore, the authors of this article decided to present an alternative method of data reduction from the experiment. Approximation by means of a set of basic functions is equally effective but easier to use.

The presented manuscript illustrates a new approach to identifying the selected parameters of the elastic–plastic material. The analysis employed the simulation results of the indentation tests by G. Maier’s research group [52]. The general algorithm of the proposed method is presented in the scheme in Figure 5. The identification process included *P-A* and *P-C* procedures. It comprised the following crucial stages:

Stage 1: The imprint indentation curves (Figure 6) and profile curves (Figure 7) used for identification consisted of 200 and 72 data, respectively. In order to facilitate the training of the neural network, i.e., to avoid overfitting and improve the ability of the general networks, both curves were approximated separately. Basic approximation functions were selected individually for individual branches of the indentation curves and curve branches describing the shape of the imprint profile. Approximation curve factors were used as input data in the next identification stage.

Stage 2: The MLP, which performs the inverse task, was developed at this stage. This network had the following architecture: 8-H-3 (Figure 8). Based on eight data describing the curves (indentation and imprints), three selected parameters of the elastic–plastic material were identified: *E*, $\sigma_{Y}$, and *n*. For the calculations, H = 15 neurons were assumed in the hidden layer of the neural network. 

In order to estimate the correctness of the results of the performed approximations (Stage 1) and NN formulation (Stage 2) (see Figure 5), the following error measures were used:


Mean squared error (MSE):
(8)$$MSE=\frac{1}{P}\sum_{p=1}^{P}\sum_{p=1}^{P}\left(t_{i}^{\left(p\right)}-y_{i}^{\left(p\right)}\right),$$
where p=1, …, P—pairs of data;$y_{i}^{\left(p\right)}$—reference data;$t_{i}^{\left(p\right)}$—computed values.Average absolute relative error (*avr ep_i_*):(9)$$MSE=\frac{1}{P}\sum_{p=1}^{P}ep_{i},\;\;\;\;\;\;\;\;\mathrm{where}\;\;\;\;\;ep_{i}=\left|1-\frac{y_{i}^{\left(p\right)}}{t_{i}^{\left(p\right)}}\right|\;\cdot \;100\%,$$Linear regression coefficient (*r_i_*): (10)$$r_{i}=\frac{\sum_{p=1}^{P}\left(t_{i}^{\left(p\right)}-{\bar{t}}_{i}\right)\left(y_{i}^{\left(p\right)}-{\bar{y}}_{i}\right)}{\sqrt{\sum_{p=1}^{P}{\left(t_{i}^{\left(p\right)}-{\bar{t}}_{i}\right)}^{2}\sum_{p=1}^{P}{\left(y_{i}^{\left(p\right)}-{\bar{y}}_{i}\right)}^{2}}},$$
where ${\bar{t}}_{i}=\frac{1}{P}\overset{P}{\underset{p=1}{\sum }}t_{i}^{\left(p\right)},{\bar{y}}_{i}=\frac{1}{P}\overset{P}{\underset{p=1}{\sum }}y_{i}^{\left(p\right)}.$


### 3.1. Stage 1 for P-A Procedure

In the approach of the *P-A* type, the material was identified based on the indentation curve. This curve shows the relationship of force F applied to the indenter from the vertical displacement of indenter *v*. Figure 6 demonstrates example indentation curves *P*(*v*) determined for different parameters of an elastic–plastic material.

Every indentation curve from the prepared set *S* = 5740 of elements is marked by 100 pseudo-measurable points (200 coordinates—Figure 6). To reduce the number of coordinates, the curve diagrams were divided into two parts, *a* and *b*, which were then approximated by the following functions:(11)$$f_{i}\left(x\right)=\Phi_{i}\left(x\right)\;a^{T}$$
where *i* = *a* or *b*; ***a***—vector of coordinates of approximating functions; $\Phi_{i}$—vector of base functions with the following form:(12)$$\Phi_{a}=\left[x,\;x^{2},\;\ln(x),\;\frac{1}{x}\right],\;\;\;\;\;\;\;\;\;\;\Phi_{b}=\left[1,\;x,\;\ln(x),\;\frac{1}{x}\right]\;.$$

Due to this approach, instead of 200 data describing each indentation curve, eight curve factors *f_i_*(*x*) were obtained, so data compression was performed. 

### 3.2. Stage 1 for P-C Procedure 

In the *P-C* approach, the identification of material parameters was performed based on the size of the imprint profile. Figure 7 shows example curves that describe the imprint profiles. These curves describe a relationship between two indenter coordinates: radial—*r* and vertical—*v*. As it is shown in Figure 7, the shape of the curve also depends on the adopted material parameters. 

Each of the set *S* = 5740 of profiles is described using 36 points (72 coordinates). In this case, curve diagrams were also separated into two parts: *c* and *d*. Branches *c* and *d* were also approximated using Formula (4), assuming *i* = *c* and *d*:(13)$$\Phi_{c}=\left[1,\;x,\;x^{2},\;x^{3}\right],\;\;\;\;\;\;\;\;\;\;\Phi_{d}=\left[1,\;x,\;\ln(x),\;\frac{1}{x}\right]\;.$$

Table 1 includes the maximum values: absolute error—*e*, mean square error—*MSE*, and linear regression coefficient—*r*, which allow for estimating the approximation quality. 

The average absolute relative error did not exceed 2.5% when approximating each branch of the graphs. The mean square errors were at an acceptable level not exceeding 1. The linear regression coefficients were close to 1, which means that Function (11) $f_{i}\left(x\right)$ reproduced the course of the coordinates describing the indentation curve and the imprint profile as accurately as possible. Based on the presented results, it can be concluded that the approximation process was conducted correctly.

### 3.3. Stage 2 for Procedures: P-A and P-C

In this manuscript, for the identification of material parameters, both in the *P-A* and *P-C* procedures, MLP was applied and developed with the use of the processed patterns generated in the indentation test. A total of 2000 patterns were selected for training the network, and 3740 patterns were adopted for testing the network. After the initial calculations, the network with the structure 8-15-3 (Figure 8) was approved for further analysis. Eight network inputs were the approximation factors, while the output from them comprised the identified material parameters (*E*, $\sigma_{Y}$*, n*). H-15 neurons in the hidden layer were adopted for calculations. Bipolar sigmoidal activation functions (*F_h_*) for the neurons of the hidden layer and linear functions in the output layer were adopted. The formulation of the network was conducted offline with the use of a Toolbox (Neural Network Toolbox [53]), working in the MATLAB computing environment [54]. The pseudo-Gaussian Levenberg–Marquardt method was used for learning.

The built MLP network 8-15-3 represented the following formula:(14)$$y\left(x;w\right)=\sum_{h=1}^{15}w_{h}^{\left(2\right)}F_{h}\left(\sum_{j=1}^{8}w_{hj}^{\left(1\right)}x_{j}+w_{0j}^{\left(1\right)}\right)+w_{0}^{\left(2\right)}\;.$$
where *F_h_*—activation functions of hidden layer neutrons; ***w***—network parameter vector (weights and biases).

MLP network preparation process: structure 8-15-3 (Figure 8) was completed after 900 training epochs for the mean square errors of learning and testing: *MSE*_L_ ≈ 1.2 × 10^−5^, *MSE*_T_ ≈ 1.1 × 10^−5^. 

Figure 9 shows the correlation of the material parameter values calculated during the NN testing and generated as a set of data used in the *P-A* identification procedure. The coefficients are $r_{E}$ = 0.924, $r_{\sigma_{Y}}$ = 0.944, and $r_{n}$ = 0.965, respectively, for the estimated Young’s modulus, yield stress, and exponent of the material strengthening degree. In the case of the *P-C* procedure, the estimation results are very similar to those above. The coefficients of regression are $r_{E}$ = 0.938, $r_{\sigma_{Y}}$ = 0.946, and $r_{n}$ = 0.954. The NN was developed correctly.

## 4. Results and Discussion

Figure 10a, Figure 11a and Figure 12a illustrate the error distributions in the estimates of Young’s modulus *E*, yield point $\sigma_{Y}$, and hardening exponent *n* obtained from the NN simulation (Stage 2). The visualized results were computed starting from “perfect data”, namely the pseudo-experimental values of the measurable quantities as provided by the preliminary direct analysis in ABAQUS. As shown in Figure 10, Figure 11 and Figure 12, the most significant errors were found by identifying the Young’s modulus E  values as 0.88% and 1.01% for *P-A* and *P-C*, respectively, and the yield point $\sigma_{Y}$ as 2.04% for the *P-A* procedure and 1.75% for the *P-C*. Figure 10b, Figure 11b and Figure 12b illustrate the error distribution for the parameter values in the estimates of Young’s modulus *E*, yield point $\sigma_{Y},$ and the hardening exponent *n* obtained from the neural network simulation (Stage 2). As can be seen, the greatest estimation errors occur by the identification of the material hardening exponent n and are approximately 20% for both types of the analyzed procedures. However, it should be noted that these errors apply to a few patterns (see Figure 12b). The analysis results indicate a similar accuracy between the *P-A* and *P-C* procedures according to the two-stage neural identification of the material parameters. Additionally, it should be mentioned that applying the proposed approach is distinguished by the higher efficiency and accuracy of the identification than the approach suggested in paper [49].

In the case of not using data compression, the inverse analysis allowing for the reading of the material data would be numerically ineffective. The neural network would have to have the following architectures: MLP: 200-H-3 for the *P-A* procedure and MLP: 72-H-3. The computing time and CPU usage would increase significantly, see [55]. The comparative assessments of the *P-A*, *P-B*, and *P-C* approaches in the works of Maier’s team [1,27] should be treated as indicative and not conclusive, as the authors define it. The above-mentioned works present the results of errors in the form of graphics, from which the maximum errors can be read approximately. For the exponent n, for example, errors comprised about 60% of single datasets in each procedure. In fact, the proposed identification procedure (compression and network errors) and the amount of experimental data may affect the comparison criterion and limit its relevance and effectiveness.

## 5. Conclusions

This manuscript exhibits the application of neural networks for identifying the parameters (*E*, $\sigma_{Y}$*, n*) of elastic–plastic material. A two-stage identification process was performed based on pseudo-experimental data. One applied two identification procedures: *P-A* and *P-C*. In the *P-A* procedure, the material parameters were identified based on the coordinates of the indentation curves, but in the *P-C* procedure, they were based on the coordinates of the imprint profile.

One of the most important scientific results of this paper is the original proposal of the input data compression of the neural network. Due to its simplicity, approximation as compression can be competitive in relation to, e.g., principal component analysis (PCA). Approximation is a very synthetic method, takes less time to solve the problem of data reduction, and requires less computation than the PCA method. The use of classic approximation allowed for reducing the architecture of the neural network from 200-*H*-3 for *P-A* and 72-*H*-3 for *P-C* to architecture 8-15-3. The analysis performed implies the following:

A neural network with back-propagation (PBNN) can be effectively used for identifying material parameters described by Ramberg–Osgood’s law, i.e., Young’s modulus E, yield point $\sigma_{Y},$ and the material hardening exponent n;A correctly performed approximation process is an effective way to reduce the multidimensionality of the input space of a neural network and allows for achieving a satisfactory accuracy in the estimation of material parameters;In the suggested approach of the two-stage hybrid identification of elastic–plastic parameters, the accuracy and parameter estimation errors of the *P-A* and *P-C* procedures are similar.

The proposed approach can be used in practice as a component of the software analyzing the results of real indentation tests. The direction of development of the presented research could be the use of the results from real indentation tests. There are also plans to investigate the possibility and effectiveness of using probabilistic models, mainly Bayesian neural networks (BNNs), as well as combining them with classic determinist NNs.

## Figures and Tables

**Figure 1 materials-15-04139-f001:**
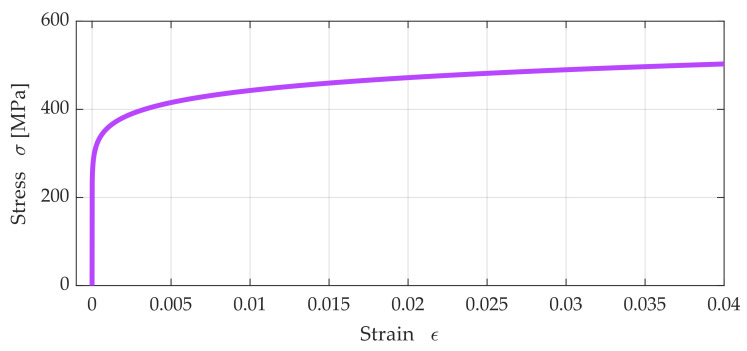
An example diagram $\sigma \left(\varepsilon \right)$ generated for elastic–plastic material with reference values: *E*_0_ = 200 GPa, $\sigma_{Y}$ = 380 MPa and *n* = 0.092.

**Figure 2 materials-15-04139-f002:**
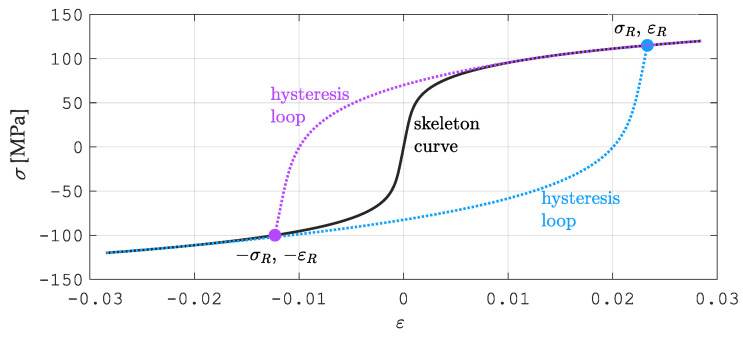
Skeleton curve and hysteresis loop for Ramberg–Osgood’s material model.

**Figure 3 materials-15-04139-f003:**
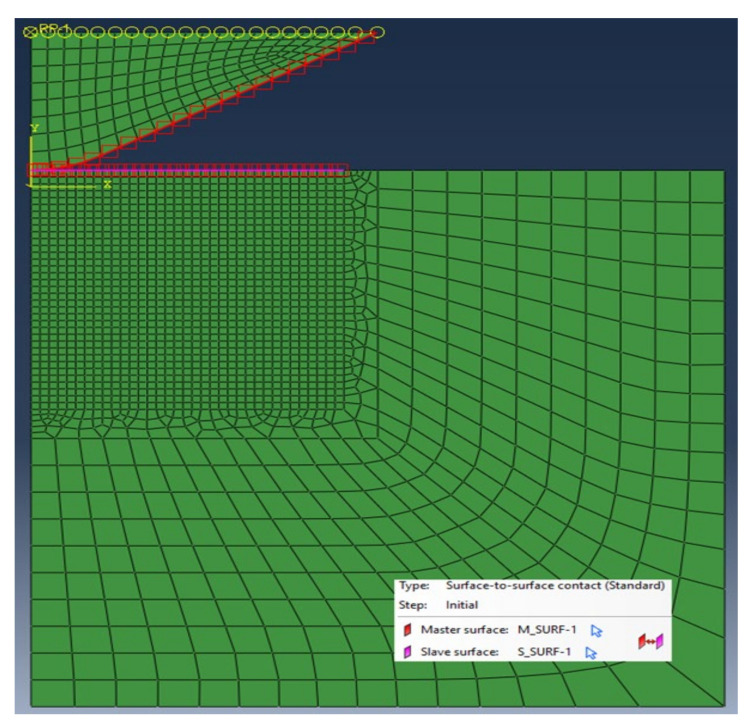
Finite element mesh and contact model of task.

**Figure 4 materials-15-04139-f004:**
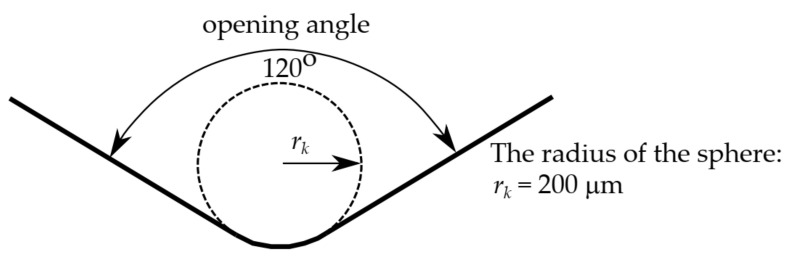
Cross-section of the tip of the sphero-conical diamond indenter.

**Figure 5 materials-15-04139-f005:**
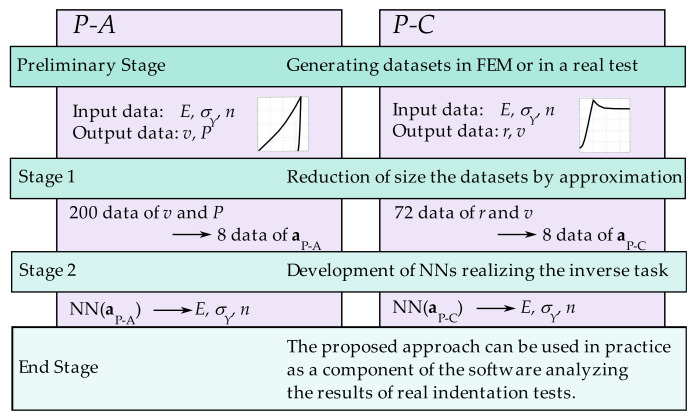
Algorithm of the applied procedure of identifying the material parameters.

**Figure 6 materials-15-04139-f006:**
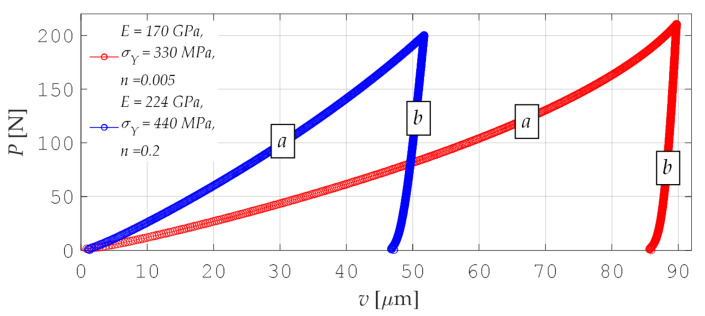
Example diagrams of indentation curves and their approximation functions (the letters *a* and *b* denote separate branches of one diagram).

**Figure 7 materials-15-04139-f007:**
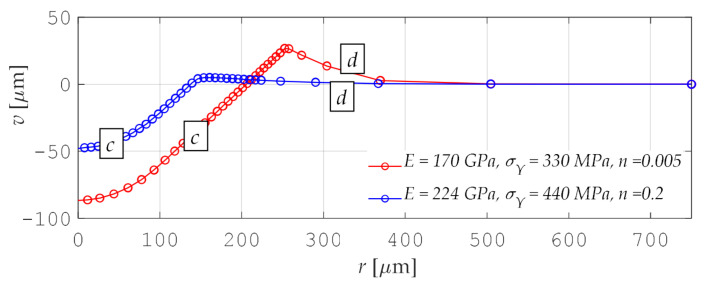
Example diagrams of imprint profiles and their approximation functions (the letters *c* and *d* denote separate branches of one diagram).

**Figure 8 materials-15-04139-f008:**
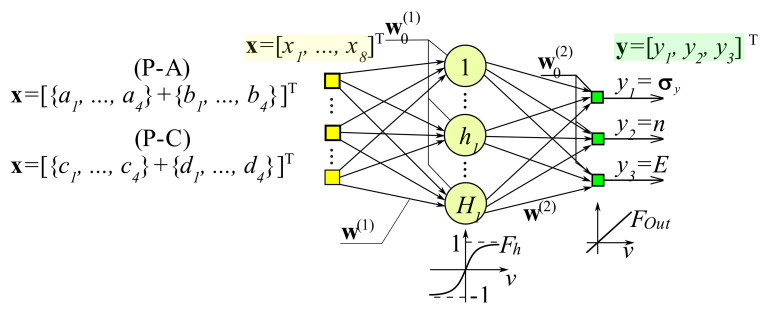
The structure of the applied MLP network.

**Figure 9 materials-15-04139-f009:**
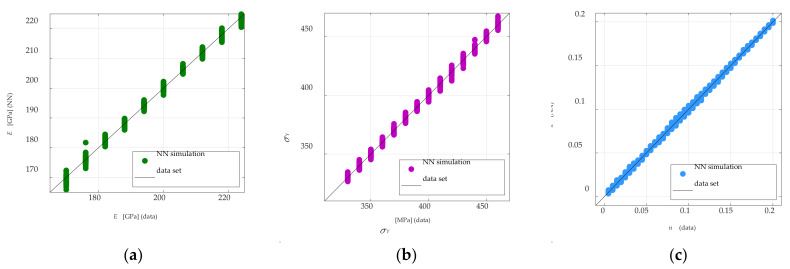
Plots for testing linear correlation in the *P-A* procedure for MLP 8-15-3; (**a**) Young’s modulus, (**b**) yield stress, (**c**) exponent of material strengthening degree.

**Figure 10 materials-15-04139-f010:**
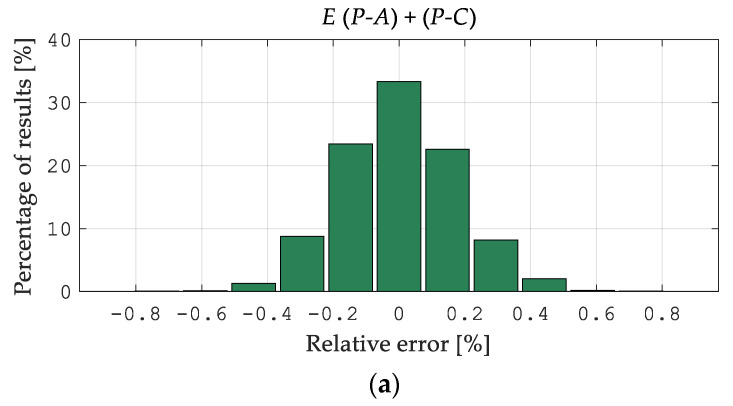
Errors of the *P-A* and *P-C* procedures in the estimates of Young’s modulus *E*: (**a**) histogram of the error distribution, (**b**) the error distribution of the parameter values.

**Figure 11 materials-15-04139-f011:**
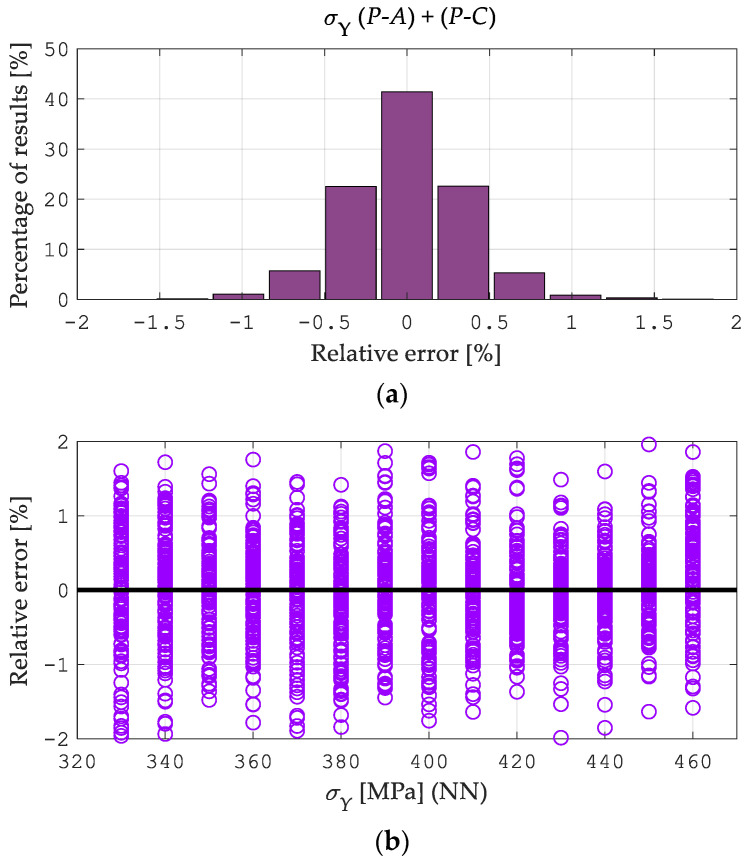
Errors of the *P-A* and *P-C* procedures in the estimates of the hardening exponent *n*: (**a**) histogram of the error distribution, (**b**) the error distribution of the parameter values.

**Figure 12 materials-15-04139-f012:**
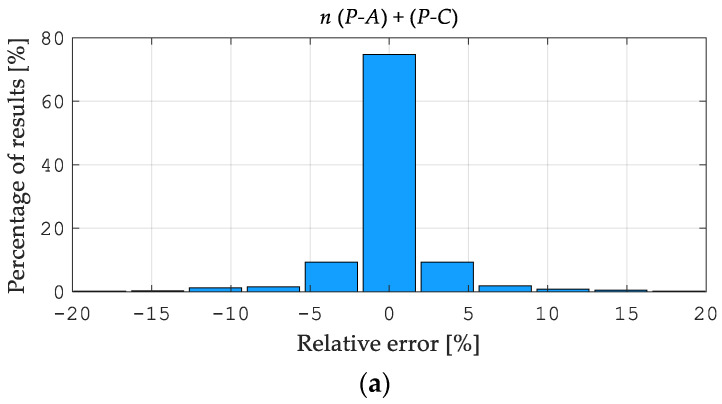
Errors of the *P-A* and *P-C* procedures in the estimates of the yield point $\sigma_{Y}$: (**a**) histogram of the error distribution, (**b**) the error distribution of the parameter values.

**Table 1 materials-15-04139-t001:** Maximum values of absolute error (*e*), mean square error (*MSE*), and linear regression coefficient (*r*) for *P-A* and *P-C* procedures.

Procedure	Curve	*avr ep* (%)	*MSE*	*r*
*P-A*	*a*	1.601	0.4751	0.9996
	*b*	0.229	0.1003	0.9982
*P-C*	*c*	2.227	0.7693	0.9989
	*d*	0.980	0.3874	0.9975

## Data Availability

The data presented in this study are available upon request from the corresponding author.

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
