# Peer review of "The Application of a Hybrid Method for the Identification of Elastic–Plastic Material Parameters"

_materials, 2022, doi:10.3390/ma15124139_

Round 1
Reviewer 1 Report
The researchers proposed a hybrid method using the finite element method (FEM) and neural network (NN) to identify material parameters. The keypoint is to compress the input data of the neural network. FEM and MLP are two very conventional methods, but the advantages of the combination were missing in this work, which is what the reviewer cares most about. In addition, the following issues are recommended to follow.
- The logic of the Introduction is confusing. The indenter introduced in the last paragraph is not related to the article's content. Until the end of this part, it did not point out the problems of the existing methods, the purpose and significance of this paper, and the whole article's framework.
- In Section 2.2, is the model equation (2) proposed by a Ramberg Osgood power-law applicable? Please provide proof. In Line 172, what does "Tri-axis version of this popular model" mean? Previously mentioned is a uniaxial model.
- This paper mentions the hybrid method of FEM and NN, but there is too little introduction about FEM. In addition, the model and data of FEM are completely from the other study.
- In Section 2.3, it is mentioned that the PCA method can reduce the amount of data. It is suggested to discuss this part in the Introduction and explain its shortcomings to put forward the compression method in this paper.
- In Line 214, "Papers [13,35] also included a comparison of identification results obtained for three approaches: P-A, P-B and P-C" was mentioned, but the specific results were not further explained.
- In Section 3.2, the title should be "Stage 1 for P-C procedure", please modify.
- In Section 4, the conclusion is only three pictures of the errors of Young's modulus, yield point and hardening exponent. The conclusion is neither compared with the effect before compression nor with PCA method.
- In Section 5, "One of the most important scientific results of this paper is an original proposal of input data compression of neural network. Due to its simplicity, approximation as a compression can be competitive in relation e.g. to Principal Component Analysis - PCA." was mentioned. What are its advantages?
- It is suggested that the text in Figure 7 should be arranged horizontally.
Reviewer 2 Report
Referee’s report
Manuscript number: 1724467
Title: Neural identification of the elastic-plastic material parameters 2 with approximation as indentation test data compression
Journal: Materials
Dear editor,
This manuscript presents a hybrid approach used to identify material parameters with the finite element method (FEM) and neural networks (NN). The suggested hybrid approach, combining both FEM and NN, effectively implements the inverse problem and can be used to identify elastic-plastic material parameters. However, this manuscript is recommended to be further refined for publication. Several critical problems need to be clarified.
- The term “finite element method (FEM)” and “neural networks (NN)” appear several times during this manuscript. The author should check the abbreviations in the manuscript.
- It’s difficult to recognize the motivation and necessity of this research with the current version of the Introduction part. The introduction should be revised.
- Please check the Ramberg-Osgood power-law equation on page #3 (Eq. 2) and their parameter’s meaning.
- In Fig. 7, the imprint for the combination of seems to be unusual, especially at the maximum depth and the accuracy of FEM of the indentation process was not illustrated.
- The methodology to determine the plastic properties of the materials from the indentation data was not presented.
- The usage of Neural Networks to determine the elastic and plastic parameters of the constitutive equation of the material was poorly presented and the estimation results should be shown in more detail.
- How does the author calculate the percentage of results in Fig. 8?
- Regarding uniqueness and sensitivity: In this paper, the author used FEM and NN to estimate both elastic and plastic parameters of the constitutive equation. It is known for years that the determination of the yield strength and strain hardening exponent using reverse finite element analysis is not straightforward. The difficulty is that several materials with different yield strength and strain hardening exponent may give identical force-displacement relationships. Please refer to the papers of (Alkorta, Martínez-Esnaola, and Gil Sevillano 2005; Nguyen, Kim, and Kim 2018; Phadikar, Bogetti, and Karlsson 2013) for detailed descriptions. The best way to overcome this problem was to use a dual indentation technique but that also suffers from a lack of reliability depending on the choice of the two tip angles. I am thus wondering about the uniqueness and sensitivity of this study. The authors have to solve this issue.
- During the indentation process, there are many factors to affect the indentation data, for example, the blunt indenter tip. Is there any methodology to reduce these factor influences to the final prediction of the proposed approach?
- The discussion section should be added.
References following the writing:
Alkorta, J., J. M. Martínez-Esnaola, and J. Gil Sevillano. 2005. “Absence of One-to-One Correspondence between Elastoplastic Properties and Sharp-Indentation Load-Penetration Data.” Journal of Materials Research 20(2):432–37. doi: 10.1557/JMR.2005.0053.
Nguyen, N. V., J. J. Kim, and S. E. Kim. 2018. “Methodology to Extract Constitutive Equation at a Strain Rate Level from Indentation Curves.” International Journal of Mechanical Sciences 152(July 2018):363–77. doi: 10.1016/J.IJMECSCI.2018.12.023.
Phadikar, J. K., T. A. Bogetti, and A. M. Karlsson. 2013. “On the Uniqueness and Sensitivity of Indentation Testing of Isotropic Materials.” International Journal of Solids and Structures 50(20–21):3242–53. doi: 10.1016/j.ijsolstr.2013.05.028.

Reviewer 3 Report
Thank you for submitting your paper. The work done here draws attention to a significant subject material properties estimation using neural networks and FE analysis. I have found the paper to be interesting. However, several issues need to be addressed properly before the paper is being considered for publication. My comments including major and minor concerns are given below:
- Please consider reviewing the abstract and highlight the novelty, major findings, and conclusions. I suggest reorganizing the abstract, highlighting the novelties introduced. The abstract should contain answers to the following questions:
- What problem was studied and why is it important?
- What methods were used?
- What conclusions can be drawn from the results? (Please provide specific results and not generic ones).
- The abstract must be improved. It should be expanded. Please use numbers or % terms to clearly shows us the results in your experimental work.
- Please consider reporting on studies related to your work from mdpi journals.
- The introduction does not read well, it is generic and should be improved, please consider improving the introduction, provide more in-depth critical review about past studies similar to your work, mention what they did and what were their main findings then highlight how does your current study brings new difference to the field.
- The authors should add a list of nomenclature for all the Greek letters and symbols used in the study.
- Line 91-95 this paragraph contains trivial knowledge which does not add any value to the manuscript, it is merely explaining what Rockwell hardness is and how it works. Please remove.
- The authors should combine all small paragraphs of 5 lines or less with other small paragraphs to improve the readability of the manuscript. Lines 98-104 is an example. Please check this issue elsewhere in the manuscript.
- Table 1 is unnecessary, the authors provide generic literature review, which is suitable for a PhD thesis, the literature review should focus on critical analysis of past studies similar to the presented work.
- Section 3.2 title can be improved.
- The title of the manuscript does not reflect the work done, the authors need to rename their manuscript to specifically mention what was done there. Most of the work is on using hardness testing to evaluate stresses and material response.
- Figure 4 should be moved to materials and method section
- I think this paper is referencing a lot of information and details from other papers such as [35]/[13]
- Line 217 what dissertation does the authors refer to? This paper seems to be cut paste from a thesis and is not properly organised/presented.
- Details about the FE model are required even though taken from other sources.
- Lines 288-298 combine in one larger paragraph.
- What are the limitation in your NN model, since error is around 20% which is quiet significant.
- The results are merely described and is limited to comparing the experimental observation and describing results. The authors are encouraged to include a more detailed results and discussion section and critically discuss the observations from this investigation with existing literature.
- Conclusion can be expanded or perhaps consider using bullet points (1-2 bullet points) from each of the subsections.
Reviewer 4 Report
REVIEW
on article
Neural identification of the elastic-plastic material parameters with approximation as indentation test data compression
Beata Potrzeszcz-Sut, Agnieszka Dudzik
SUMMARY
The study is devoted to the actual problem of identifying the elastic-plastic properties of materials by continuous indentation of a conical indenter. A hybrid approach is presented for determining material properties using the Finite Element Method (FEM) and Neural Networks (NN). To train and test the neural network, pseudo-experimental datasets were used, including indentation curves and indentation profiles obtained by modeling the indentation test using the FEM Abaqus/CAE program.
In the study, the authors used data compression to reduce the amount of input to the neural network, avoid over-fitting, and improve the ability of generalizing networks. For this purpose, the approximation of each branch of the obtained curves was used. For identification, a multilayer neural network Multilayer Perceptron (MLP) with 8-15-3 architecture was used.
The authors summarize that a backpropagation neural network (PBNN) can be effectively used to determine the material parameters described by the Ramberg-Osgood law, i.e. Young's modulus, yield strength and hardening index of the material; a correctly performed approximation process is an effective way to reduce the multidimensionality of the input space of a neural network and allows achieving a satisfactory accuracy of estimating the material parameters.
The practical results obtained are useful and can be applied in practice to determine and predict the parameters of an elastic-plastic material with satisfactory accuracy. The article is of interest to readers.
However, the article still needs to be improved. For publication of the manuscript, it is necessary to correct the comments below.
COMMENTS
- The Abstract needs to be rewritten, since in current form it is a listing of the methods and techniques used in the study. The statement of the problem that the study solves, the purpose, and the main results, including the accuracy of the forecast, should be added.
- The "Introduction" section is not logically completed and does not have a transition to the next section "Materials and Methods". As presented, these two sections are poorly interconnected. It is necessary to complete Section 1 with the formulation of the goal, the task of the study, as well as the scientific novelty that the manuscript carries.
- The presentation of Figure 1 and its description in the "Introduction" section are not entirely justified. Perhaps it should be moved to the next section on methods and materials.
- In general, given that out of 17 references in the "Introduction" section, only 14 are references to the works of other authors, another 5-10 fresh literature sources published no earlier than 2017 should be added. For example, Materials 2020, 13, 2445; doi:10.3390/ma13112445. Thus, this would make it possible to formulate more accurately and succinctly what was the scientific deficit and the need for the ongoing research.
- In the second section "Materials and Methods", Table 1 should have been moved immediately after the first mention of it in the text (line 109).
- The use of pseudo-experimental data obtained based on the FEM analysis is doubtful. Thus, you coarsen all the richness of real patterns and replace them with a deterministic data set.
- Equation (8) Is duplicated twice - some kind of mistake.
- It is necessary to add in the text a mention of figure 2 before it.
- Subsection 2.2 ends with Figure 3. It is necessary to improve the readability of this subsection and figure by adding a brief interpretation of Figure 3 after it.
- The same remark applies to subsection 2.3 and figure 5. It is necessary to link sections 2 and 3 more smoothly by adding an interpretation of figure 5 after the figure itself and, perhaps, brief conclusions in section 2.
- The name of subsection 2.3 "Inspiration" is not entirely clear. It might be worth clarifying this heading choice by adding information about it at the end of subsection 2.3. According to the reviewer, in this form, the title and the content of the subsection have little connection.
- Perhaps line 249 should mention Figure 7 rather than Figure 5. Should be checked and corrected if necessary.
- Subsection 3.2 ends with the sentence at lines 264, 265 "Based on the presented results, it can be concluded that the approximation process was conducted correctly". It is necessary to substantiate this conclusion in more detail.
- It is necessary to correct the numbering of figures in the manuscript. In its present form, the manuscript contains two drawings 7 on lines 253 and 280.
- In line 288, instead of figure 7, perhaps figure 8 was meant. It should be checked and, if necessary, corrected.
- Section 4 is rather concise. It would be necessary to strengthen this section with a more detailed description of Figure 8, as well as add a few paragraphs of discussion - a more detailed comparison of the methods proposed by the authors with existing solutions. This section needs improvement.
- It is not clear how accurate the property predictive value is.
- Section 5 should be supplemented with information on the practical application of the results obtained and the prospects for further research in the chosen direction.
- The "References" section must be edited in accordance with the established requirements for the design of the list of references. Attention is drawn to the difference in styles of design of sources of the same type. I also recommend adding a DOI for faster browsing of articles.
In general, the article is devoted to an important and topical topic of determining the mechanical characteristics of materials and will undoubtedly attract the attention of readers.
Round 2
Reviewer 1 Report
Sufficient revisions have been made by the authors. The paper can be published in its current form.
Author Response
Thank you for your positive review No 2.We were very pleased to respond to your comments.
Reviewer 2 Report
English language and style are fine/minor spell check is required.
Author Response

(The authors gave the same response as above.)

Reviewer 3 Report
Abstract still does not read well. Please improve and list main and specific findings, not generic ones.
Section 2.4 is not adding any value, very generic and does not answer the questiosn raised in previous round of review.
Authors are encouraged to use standard paper format, for example section 3 use Results and discuss
What is the percentage of self citation in this article?
Reviewer 4 Report
All my comments were considered and appropriate corrections were made in the article's text. The article looks much better. I recommend the article for publication.
Author Response
Thank you for your positive review No 2.
We were very pleased to respond to your comments.